# Optimal distribution-preserving downsampling of large biomedical data sets (opdisDownsampling)

**Jörn Lötsch**[1,2]*, **Sebastian Malkusch**[1], **Alfred Ultsch**[3]

**1** Institute of Clinical Pharmacology, Goethe—University, Frankfurt am Main, Germany, **2** Fraunhofer Institute for Translational Medicine and Pharmacology ITMP, Frankfurt am Main, Germany, **3** DataBionics Research Group, University of Marburg, Marburg, Germany

* j.loetsch@em.uni-frankfurt.de

**Data Availability Statement:** The R package "opdisDownsampling" is freely available at https://cran.r-project.org/package=opdisDownsampling, which contains data sets #3 and #5 used in this report; the other data sets are available from

## Abstract

### Motivation

The size of today's biomedical data sets pushes computer equipment to its limits, even for seemingly standard analysis tasks such as data projection or clustering. Reducing large biomedical data by downsampling is therefore a common early step in data processing, often performed as random uniform class-proportional downsampling. In this report, we hypothesized that this can be optimized to obtain samples that better reflect the entire data set than those obtained using the current standard method.

### Results

By repeating the random sampling and comparing the distribution of the drawn sample with the distribution of the original data, it was possible to establish a method for obtaining subsets of data that better reflect the entire data set than taking only the first randomly selected subsample, as is the current standard. Experiments on artificial and real biomedical data sets showed that the reconstruction of the remaining data from the original data set from the downsampled data improved significantly. This was observed with both principal component analysis and autoencoding neural networks. The fidelity was dependent on both the number of cases drawn from the original and the number of samples drawn.

### Conclusions

Optimal distribution-preserving class-proportional downsampling yields data subsets that reflect the structure of the entire data better than those obtained with the standard method. By using distributional similarity as the only selection criterion, the proposed method does not in any way affect the results of a later planned analysis.

sources indicated with their description in this report.

**Funding:** This work has been funded by the Landesoffensive zur Entwicklung wissenschaftlich-ökonomischer Exzellenz (LOEWE), LOEWE-Zentrum für Translationale Medizin und Pharmakologie (JL), in particular through the project "Reproducible cleaning of biomedical laboratory data using methods of visualization, error correction and transformation implemented as interactive R-notebooks" (JL). The funders had no role in the decision to publish or in the preparation of the manuscript.

**Competing interests:** The authors have declared that no further conflicts of interest exist.

# Introduction

With the development of biomedical research over the past two decades, data sets are becoming increasingly larger. This has been accompanied by a move toward increasingly powerful computing facilities [1]. Nevertheless, the number of operations performed in the analysis of large biomedical data sets and the size of the data stored can easily exceed the capacity of today's computers. This can happen even with seemingly simple tasks such as data projection for visualization. The number of unique distances, $n_{dist}$, between data points that need to be calculated for this task is proportional to the square of the number of instances; precisely, $n_{dist} = O(n^2) = \frac{n^2 - n}{2}$. For 150 data points, this gives 11,175 distance values; however, for 100,000 data points, the number raises to 4,999,950,000 (4.99995 billion). To store these distances in a vector of 64-bit numeric units in a computer's memory, for example, using the R software environment, 37.25287 gigabytes (GB) are needed [2]. This is only for one variable. Already for a data set with $1,000,000 = 10^6$ instances and 10 variables and a projection using multidimensional scaling (MDS), which requires the full distance matrix of size n², the $10^{12}$ numbers require 7450.6 GB of memory to be stored during the calculations. Typical biomedical data sets nowadays, e.g., from flow cytometry, often contain many variables with $10^6$ or more instances (big data).

Reducing the number of instances (downsampling) in a data set is therefore a common step in the analysis of big biomedical data. Downsampling approaches include simple random sampling where each case of the population has the same chance of being selected for the sample. In this sampling strategy the probability of occurrence of a case in the reduced data set is direct proportional to the probability of occurrence of that case in the full data set. Alternatively, sampling strategies can involve some form of selective filtering as in so-called gating approaches, where, for example, more frequent instances in the original data set are preferentially selected for the reduced data set. The rationale for this is that biomedical data sets may contain noise which blurs the separation of disease-related subgroups in the data. This may be true for flow cytometric data sets where the focus is on specific cell populations that are initially separated from cell types that are not of interest, but criteria for preselecting specific parts of any large data set are not always available without anticipating future results.

Uniform, i.e., simple random sampling procedures are part of most statistical software packages. This is the focus of this report, which proposes an improvement to the uniform downsampling method. Uniform downsampling implies that each instance of a data set has the same chance to be included in the reduced data set, which equals to a probability $p_{sampled}$ for the occurrence of a case in the sample of $p_{sampled} = \frac{n_{sampled}}{n_{total}}$. However, there are many possible combinations for the exact composition of the down sampled data set from the available instances. This complexity increases when the data set contains a priori classes that should be represented in the reduced data set in proportional to its original size (also known as stratified or quoted sampling). It is reasonable to assume that an arbitrary randomly drawn subset of the data will represent the original data set, although eventually not really well. Therefore, the hypothesis pursued in this report was that class-proportional, uniformly distributed downsampling of data sets can be optimized to obtain samples of data that better reflect the entire data set, rather than taking only the first randomly drawn subsample, which is the usual approach. Here, sample selection among many randomly drawn samples was based on the similarity of the distribution in each variable of the down sampled variables to the respective distribution in the full data.

## Methods

### Development

**Basic considerations.** The number of possible combinations of $r$ instances of a data set with $n$ cases is given by the binomial coefficient $\binom{n}{r}$. If a fraction of size f with $f \in [0,1]$ is drawn from a data set and the data set contains k > 1 classes, this is true for each class and, as stated in the S1 File of this report, if the sampling is class-proportional, i.e., keeping the priors of the classes, this potentially leads to a huge number of different combinations of instances that can be drawn. For example, from the famous Iris data set (data set #2, see next subchapter [3, 4]), which contains three classes with each 50 instances, sampling of none or 100% of the data leads to one single combination. Sampling of f = 90% of the data implies already $9.5114186 \cdot 10^{18}$ (trillions) possible combinations. A maximum of possible combinations is obtained when f = 50% of the data is sampled of $2.019996165 \cdot 10^{42}$ (septillions) possible combinations. The general relation between the number of possible combinations and the fraction sampled from the data set following a bell-shaped curve based on the theoretical probabilities shown in the S1 File of this report.

Thus, as mentioned in the introduction, if a very small subset $f \ll 1$ is uniformly sampled from big data, i.e., each instance of the original data set has an equal chance of being part of the downsampled data set, the particular composition of instances in the sample drawn is highly random. The present approach was based on the hypothesis that, given the large number of possible combinations of instances that can be encountered in a class-proportional downsampling of a data set, the sample drawn first may not be representative of the full data set (**Fig 1**). By repeating the sampling, it is possible to obtain samples that reflect the entire data set better or worse. It was also hypothesized that the improvement of the common sampling strategy does not require exhaustive search among all possible combinations, which would be impossible or unrealistic. It was expected that even with reasonably sized searches among 1,000 or 10,000 trials it should be possible to obtain a subsample that reflects the original structure comparatively well.

**Criterion for sample selection.** The criterion by which a particular sample is selected from several possible uniform sample combinations should not be biased by any a priori hypothesis on the data and in particular on possible groups. It should be derived solely from the similarity of the distribution of the sample to the distribution of the full data set (see also https://en.wikipedia.org/wiki/Loss_function). Therefore, the similarity of distributions between the sampled data and the original data was chosen as the target criterion for sample selection. More specifically the (dis-)similarity of the probability density distributions (PDF) respectively of the cumulative distribution function (CDF) was used. For numerical calculations 200 kernels within the data's range were used. For the calculation of the PDF at the kernels the Smoothed Data Histograms (SDH [5]) with default parameter settings was used. Several methods for comparing density distributions are available, including various methods used in statistical tests and other distance measures on vectorial data. Several of such distance metrics were evaluated. The distance metrics comprised, in alphabetical order, (i) the absolute mean relative difference of the distributions of the subsample and of the original variable,

$$AbsMeanRelDiff = \frac{\sum_i^n \left| \frac{P_{sample,i} - P_{Original,i}}{0.5 \cdot (P_{sample,i} + P_{Original,i})} \right|}{n}$$

[6], (ii) the Anderson-Darling test [7], (iii) the Cramér-von Mises test [8, 9], (iv) the empirical cumulative distribution function ECDF two-sample test [10], (v) the Euclidean distance on the PDF, (vi) the Kolmogorov-Smirnov test [11], (vii) the Kuiper test [12], (viii) the symmetrized Kullback-Leibler divergence [13] for the PDF

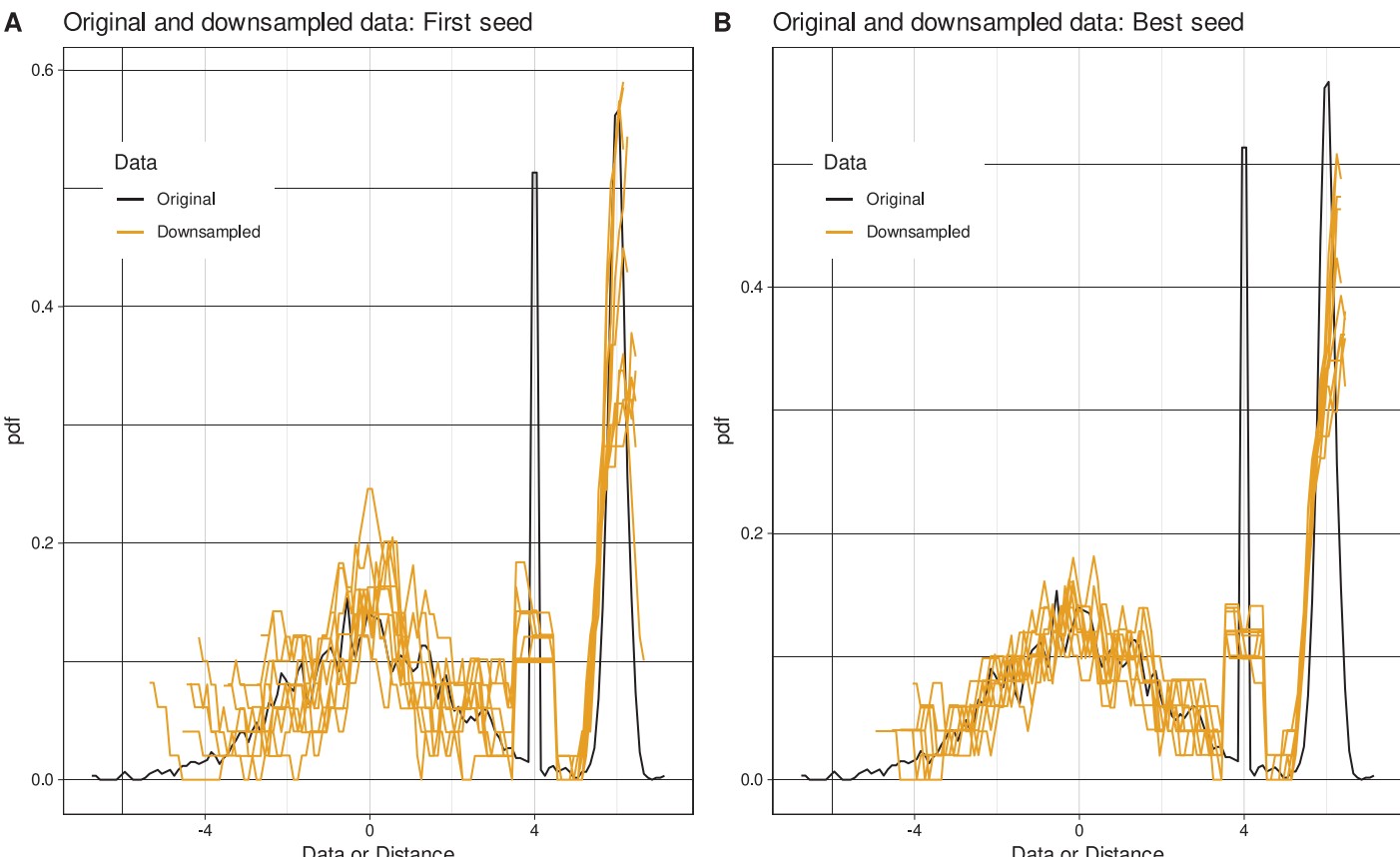

**Fig 1. Distribution of 50 data points drawn 10 times uniformly from an original artificial data set of 3,000 instances, using either classical uniform sampling or the optimized sampling method.** The original data included 3,000 instances from three classes with prior probabilities of [0.6, 0.1, 0.3]. The distributions of cases within the classes follow Gaussian distributions with means = [0, 4, 6] and standard deviations = [2, 0.001, 0.2]. The panels show the probability density distribution of the original data (black lines) and the sampled subsets (yellow lines). The probability density distribution is estimated using Pareto Density Estimation (PDE [32]). PDE is a kernel density estimator designed to be particularly useful for detecting classes [32]. **A:** 10 downsampled data subsets when using just the first of 10 different seeds. **B:** 10 optimal downsampled data subsets identified from 1,000,000 non-redundant seeds each. The figure has been created using the R software package (version 4.0.4 for Linux; https://CRAN.R-project.org/ [2]) and the R libraries "ggplot2" (https://cran.r-project.org/package=ggplot2 [33]), "ggthemes" (https://cran.r-project.org/package= ggthemes [34]) and "AdaptGauss" (https://cran.r-project.org/package=AdaptGauss [35]).

and (ix) the Wasserstein distance based two-sample test [14]. Using these metrics, the best data subsample, among many uniformly drawn random samples, was defined as the one that had the smallest distance in each variable from the distribution of the respective original variable, hence

$$BestSample = \min(\max(Distance(Sample - Original))),$$

where "Distance" denotes one of the metrics mentioned above.

To select an appropriate metric for choosing the best sample in terms of the similar distribution of the subsampled data and the original data, the Iris data set (see below) was used to compare the quality of the reconstructed data sets under several statistical tests of distribution similarity. Assessments were performed in three steps, consisting of (i) samples of size 1% or 10% of the original data either drawn once or repeatedly 10,000 times, (ii) selecting the most appropriate sample based on the similarity of the PDF of the sample to the PDF of the full sample, and (iii) using a PCA projection of this sample as described below to reconstruct the rest of the iris data set.

## Evaluation

The programming work for this report was performed in the R language [15] using the R software package [2] (version 4.0.4 for Linux), which is available free of charge in the Comprehensive R Archive Network (CRAN) at https://CRAN.R-project.org/ [2].

**Data sets.** The **first data set** was created for problem introduction and simply consisted of samples from three classes with priors [0.6, 0.1, 0.3]. The distribution of cases within the classes follows Gaussian distributions with respective means = [0, 4, 6] and standard deviations = [2, 0.001, 0.2]. 1,000 instances for each class, i.e., 3000 samples in total, were randomly generated for this data set.

The **second data set** was the iris flower data set [3, 4], which has been used for method development in statistics when Fisher introduced linear discriminant analysis [3]. The Iris data has been widely used in statistics for testing of methods. It gives the measurements in centimeters of the four variables sepal length and width or petal length and width for 50 flowers each of the three species *Iris setosa*, *versicolor* and *virginica*. As there are apparently at least half a dozen different versions of this data set [16], it is necessary to specify that in the present analysis, the version implemented in R software package as "data(iris)" was used.

The **third data set** contained artificial data with a clear structure consisting of 10 variables with a three-modal distribution corresponding to three groups to which the 30,000 instances belonged. Specifically, the data set was created by randomly generating Gaussian distributions with priors [0.5, 0.4, 0.1]. The means of the Gaussians were uniform randomly selected from a range of [1,. . .,100], with a uniform selected standard deviation within the range [1,. . .,30]. Group numbers were assigned in ascending order of the selected means. The data set is provided with the R library described below that accompanies this report.

The **fourth data set** was a real live single-cell flow cytometry data set used to as benchmark data set in the development of a clustering algorithm for FACS data [17]. This was the sample data provided with the software presented in the cited paper and is freely available at https://github.com/VCCRI/FlowGrid (downloaded February 5, 2021). The data set contains four markers (FL1.H, FL2.H, FL3.H, FL4.H) measured for 23,377 cells. Samples are labeled according to four different samples of sizes 453, 17,467, 929, and 4,528.

The **fifth data set** was a real live data set consisting of empirical data from flow cytometry by fluorescence activated cell sorting (FACS). The distribution of the variables in flow cytometry data is usually complex. The present sample data set included n = 111,686 cells obtained from 100 patients with chronic lymphocytic leukemia and 100 healthy controls, comprising d = 6 cytological markers which for privacy protection are not disclosed here.

The **sixth data set** was again of biomedical origin and consisted of a "miRNA and chronic pain" data set containing measurements of 184 microRNAs in 94 instances grouped into d = 5 classes of sizes [14, 24, 18, 17, 21]. The data set is freely available under the Creative Commons license CC By 4.0 at https://data.mendeley.com/datasets/37fnjc4yhm/2 (downloaded at June 14, 2021). Of its two versions, the "Profiling raw data.xlsx" file dated February 19, 2020 was selected [18].

**Performance measure.** The efficiency performance of the downsampling method in terms of drawing a most typical sample was measured by the mean square error (MSE) with which the remaining data in the data set can be reconstructed from the downsampled data.

**Data reconstruction.** In all experiments, the correctness of the prediction of the remaining data from the downsampled data was assessed as follows. The downsampled data $X_{sample}$ were projected and this projection was used to reconstruct the remaining data ($X_{recosample}$).

Thus, the reconstruction-MSE calculates to:

$$MSE = \frac{\sum_{i=1}^{n} \sum_{j=1}^{p} \left( \boldsymbol{X}_{recosample}[i,j] - \boldsymbol{X}_{remaining}[i,j] \right)^2}{n \, p}$$

Where $X_{remaining}$ denotes the original data without $X_{sample}$, n denotes the number of unselected instances, and p denotes the number of features in the data set. If this resulted in smaller errors than a randomly selected subsample, it was concluded that the downsampled data better reflected the structure of the original data set.

Principal component analysis (PCA [19, 20]) was chosen as a suitable projection method that could be easily reversed to reconstruct the original data. PCA was performed on non-scaled and centered providing an orthogonal projection of the data onto a low-dimensional linear space named the principal subspace. The first factor of the PCA is the direction of greatest variance in the data. As principal those components (PC) were selected which passed the Kaiser-Guttman criterion [21, 22], i.e., PCs which correspond to eigenvalues > 1 of the covariance matrix. The PCA results were then used to predict the other data in the original data set ("RemovedData") that were not included in the downsampled data subset. Specifically, the projected data were transformed back from the low-dimensional space into their original coordinate space using standard procedures that for convenience are specified in the S1 File of this report. Of note, using all PCs instead of PCs with eigenvalues > 1 did not result in any relevant changes to the observations and conclusions reported here.

**Experimentation.** We performed all experiments on 1–8 cores of an Intel Core i9-7940X® (Intel Corporation, Santa Clara, CA, USA) computer with 128 GB Random-Access Memory (RAM) running Ubuntu Linux 20.04.2 LTS (Canonical, London, UK)). To reproduce all the comparisons presented in this paper, the source code and data can be downloaded from the Comprehensive R Archive Network (CRAN) at https://cran.r-project.org/package=opdisDownsampling.

The experiments were designed to evaluate whether a sample selected by the proposed method better represents the original data set than an initial random sample. The experiments focused on (i) comparing measures of distributional similarity between downsampled and original data, (ii) evaluating data reconstruction errors after distribution-preserving downsampling, and (iii) internally validating the consequences of distribution-preserving downsampling for data reconstruction.

*Comparison of measures of distribution similarity between downsampled and original data.* Several alternative choices of distribution similarity metrics were sequentially tested to compare their suitability in for the present method. This comprised the tests and metrics mentioned above in the method definition section of the chapter. For this test, the Iris data set was used due to its computational speed and wide use in method development. The values represent the means and standard deviations of the mean square errors (MSE) of PCA-based reproduction of the remaining data from the downsampled data. The experiments were performed in 20 replicates starting with different and non-redundant seeds, and the means and standard deviations of the mean square errors of the data reproduction obtained during these replicates were calculated. The similarity measures were ranked with respect to the MSE obtained when the final sample is chosen from 10,000 random samples. For comparison, the experiments were also done using just the first random sample from different and non-redundant seeds.

*Evaluation of data reconstruction errors after distribution-preserving downsampling.* After selecting a metric on which to base sample selection among many random samples, the downsampling experiments were conducted in data sets #2 - #5. The one-dimensional data set #1 was not included because it had served only as an introductory example. Depending on the size of the data sets, class-proportional uniform random samples of, e.g., 0.001, 0.01, 0.1, 1, 5,

10, 25, and 50% of the original data were drawn. Subsequently, the sampled data were projected using PCA onto its first principal components. The parameters of this projection were used as described above to predict the remaining data not from the original data set. The experiments were performed in 20 replicates starting with different and non-redundant seeds.

The obtained mean squared errors of this reconstruction were analyzed for a relationship with (i) the size of the fraction sampled from the original data sets and (ii) the number of trials of random class-proportional downsampling from which the final sample could be selected. For this purpose, the MSE values were subjected an analysis of variance (ANOVA) with the factors "fraction" and "number of trials". The calculations were performed with the R basic command "aov" and the α-level was set at 0.05.

*Internal validation of the consequences of distribution-preserving downsampling for data reconstruction.* The evaluation of the quality of the prediction of the remaining data from the downsampled data in different resampling scenarios was repeated by replacing the PCA-based data reconstruction with autoencoders. That is, as used elsewhere in a biomedical context [23], autoencoders use supervised learning multilayer feedforward artificial neural networks (ANN) to extract the essential features of the structure of a data set, which reduces its dimensions and thus can be used for data projection. Autoencoders then learn to reconstruct the original data with the reduced representation. The training of an autoencoder artificial neural network is done with the goal of "identity", i.e., all case vectors used as input for the autoencoder are identically reproduced as its output [24]. The neurons compute the logistic sigmoid function applied to the scalar product of the preceding neurons and the intervening synoptic weights. The common backpropagation method was used for the adaptation (learning) of the synoptic weights [25]. The network consisted of as many input and output neurons as features in the data set. After a grid search with 1 or 3 hidden layers with different sizes, a single hidden layer with 30 neurons was chosen as best performing for the present data sets. Further tuning was intentionally not performed to ensure that the downsampling method proposed here was being evaluated rather than the tuning success of the neural network, i.e., that all comparisons were performed under the same conditions. These calculations were performed using the R library "ANN2" (https://cran.r-project.org/package=ANN2 [26]). Again, the experiments were performed in 20 replicates starting with different and non-redundant seeds.

## Implementation

The method of optimized distribution-preserving data downsampling has been implemented in the R library "opdisDownsampling" freely available the Comprehensive R Archive Network (CRAN) at https://cran.r-project.org/package=opdisDownsampling. The process can be called the with the function "opdisDownsampling(Data, Cls, Size, Seed, nTrials, TestStat = "ad", Max-Cores = 8, PCAimportance = FALSE)". The input (i) is expected to include a data vector or matrix ("Data") and (ii) the total number of instances of the downsampled data set ("Size"), which must be less than the number of instances in the original data set. In addition, (iii) the class information per instance can be entered ("Cls"), if available, which provides the basis for class-proportional downsampling and is implemented using the "sample.split" function of the R library "caTools" (https://cran.r-project.org/package=caTools [27]). Other parameters include (iv) a predefined seed ("Seed") that can be used to control downsampling and can also be used to combine multiple downsampling runs into a larger overall run. Internally, the program uses seeds of [1,. . .,nTrials]. The number of trials can be set with (v) the parameter "nTrials, which is set to 1,000 by default.

Furthermore, (vi) the parameter "TestStat" can be set to select the statistical criterion for the similarity assessment between the distributions of downsampled and original variables. Because

the present comparative analyses had pointed to the Anderson-Darling test, this was set as the standard ("ad"). Further tests are mainly imported from the R library "twosamples" (https://CRAN.R-project.org/package=twosamples [28]) and consist of the Kuiper test ("kuiper"), the Cramér-von Mises test ("cvm"), the Wasserstein test ("wass"), the ECDF two-sample test ("dts"), and in addition the Kolmogorov-Smirnov test ("ks") taken from the R base environment, the symmetrized Kullback-Leibler divergence ("kld"), the Euclidean distance ("euc") and the absolute mean relative difference ("amrdd") between the two probability density distributions. PDFs were calculated at 200 kernels within the data's range using SDH kernel estimation [5].

Optionally, the library also provides (vii) a fast PCA-based feature selection ("PCAimportance") to exclude variables from the assessment of similarity to the distributions of the downsampled data and the original data to avoid unnecessarily optimizing the sample for irrelevant variables. This feature is disabled by default. The standard "prcomp" method implemented in base R [2] is used, and variables' selection was based the loadings on relevant principal components according to the Kaiser-Gutman criterion, i.e., on PCs with eigenvalues > 1. Specifically, the relevant variables are selected as suggested in the R library "factoextra" (https://cran.r-project.org/package=factoextra [29, 30]) based on the expected value if the contributions were uniform, which is given as 100/length(contrib) with "contrib" denoting the list of contribution magnitudes of each variable to a given PC.

On Unix-like systems, the calculations are performed using parallel processing depending on the computer hardware and actual data set size. This uses the "parallel" library provided with the R base environment and in addition for drawing progress bars the R library "pbmcapply"(https://CRAN.R-project.org/package=pbmcapply [31]). The number of cores can be limited using (vii) the parameter "MaxCores". Depending on the limitations of the computer's memory, the process can be divided into smaller portions that are processed sequentially, reducing the RAM requirements for a single run, using the parameter "JobSize". For fast processing, the default is set to 10,000 simultaneously executed trials.

## Results

The analyses supported the hypothesis that the generic method implemented in many data analysis programs, of using only the first result of a class-proportional uniform sample from a larger data set, may not represent the structure of the original data set well. By repeating the sampling and comparing the distribution of the drawn sample to the distribution of the original data, it was possible to establish a method for obtaining subsets of data that, on average, better reflect the entire data set than taking only the first randomly selected subsample.

A general observation during the experiments was that the reconstruction of the remaining data from the original data set using the downsampled data improved as the size of the downsampled data subset increased. That a larger portion of the data set better reflected the entire data set was an expected behavior and was taken as a positive control that the chosen strategy of PCA-based reconstruction of the remaining data from the downsampled subset was an appropriate measure of the degree to which the entire data set was represented in a subset.

Another consistent observation was that the reconstruction error decreased as the number of sampling trials increased, among which the most representative data subset was chosen based on the similarity of the distributions of each variable between the downsampled and original data.

### Comparative suitability of measures of distribution similarity for sample selection

PCA-based reconstruction of the remaining data from the downsampled Iris data set yielded smaller MSEs for 10% of the original data that were downsampled than for 1% that were

**Table 1. Comparison of various tests for differences between the distribution of the downsampled data compared to the distribution of the full data set.** For this test, the Iris data set was used due to its computational speed and wide use in method development. The values represent the means and standard deviations of the mean square errors (MSE) of PCA-based reproduction of the remaining data from the downsampled data. The experiments were performed in 20 replicates starting with different seeds, and the means and standard deviations of the mean square errors of the data reproduction obtained during these replicates are shown. The similarity measures are sorted in ascending order by the ranks of the MSE obtained when the final sample is chosen from 10,000 random samples.

| Distance test | 1% sampled 1 trial | 1% sampled 10,000 trials | 10% sampled 1 trial | 10% sampled 10,000 trials |
|---|---|---|---|---|
| Anderson-Darling | 0.14180 ± 0.04654 | 0.09271 ± 0.00524 | 0.09951 ± 0.00901 | 0.08942 ± 0.00223 |
| Cramér-von Mises | | 0.09608 ± 0.00384 | | 0.08903 ± 0.00274 |
| Kolmogorov-Smirnov | | 0.09408 ± 0.00300 | | 0.09042 ± 0.00196 |
| Mean absolute relative density difference | | 0.04215 ± 0.00717 | | 0.09590 ± 0.00989 |
| Kuiper | | 0.11496 ± 0.01739 | | 0.09037 ± 0.00334 |
| ECDF Two-Sample Test | | 0.10991 ± 0.00956 | | 0.09245 ± 0.00322 |
| Wasserstein | | 0.11394 ± 0.01579 | | 0.09060 ± 0.00318 |
| Kullback-Leibler divergence | | 0.10814 ± 0.00735 | | 0.10232 ± 0.00991 |
| Euclidean distance | | 0.12021 ± 0.01755 | | 0.09332 ± 0.00640 |

downsampled, and decreased on average when the final sample was selected from 10,000 samples than when the first sample drawn was used (**Table 1**). As expected, when only the first sample was used, it became irrelevant which distance measure was used to compare its distribution with that of the entire data set, and the reconstruction error was always the same. This was implemented as a positive check that the approach was working correctly. The smallest reconstruction-MSEs with both 1% and 10% downsampled data were obtained when assessing the similarity of the variable distribution using the Anderson-Darling test, which was therefore selected as the default for subsequent test scenarios.

## Data reconstruction errors after distribution-preserving downsampling

Depending on the original sizes of the data sets, downsampling, and subsequent reconstruction of the remaining data from the PCA projection of the sampled subset, the effects of distribution preservation reached a maximum at a certain size of the subsample relative to the full size of the data set, beyond which reconstruction errors stagnated. In all data sets (#2 - #5), the MSEs with which the remaining data could be reconstructed from the projection of the sampled data decreased (i) with increasing relative subsample size and (ii) with increasing number of samples from which the final subsample was selected (**Fig 2**). In 20 repetitions of the experiments, each staring at different and non-overlapping seed, this was reflected in significant ANOVA main effects of the factors "fraction" and "number of trials" (**Table 2**). In addition, significant interactions between the two factors indicated that the reduction in reconstruction errors achieved by selecting the most similarly distributed sample was more pronounced when reconstruction was more complicated due to the small size of the subsample data. This also captured the observation mentioned above that from a certain sample size, errors did not decrease further when using class-proportionally optimized downsampling.

## Validation of the improvement of data reconstruction using distribution-preserving downsampling

The consistent trend observed across all data sets toward smaller MSEs of PCA-based reconstruction of the remaining data from the downsampled subset (i) with increasing fraction of sampled data from the original data set and (ii) with increasing number of trial samples from which the best sample could be selected was also observed for data reconstruction with artificial neural networks (**Table 3**). That is, better agreement of patterns with the original data in

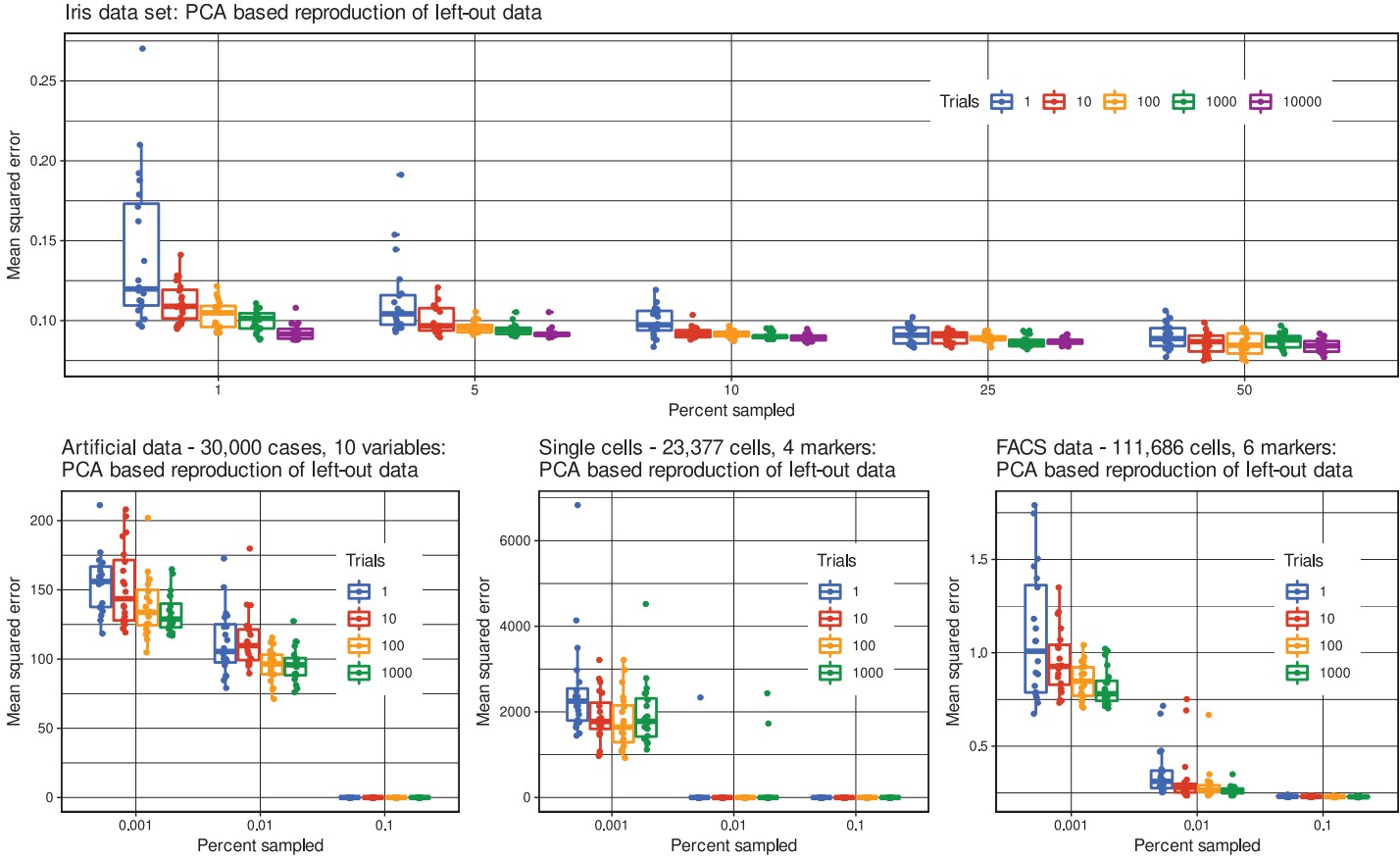

**Fig 2. Mean squared errors of PCA -based data reproduction of the remaining data from the sampled data subset.** Samples of 0.001 to 50% of the original data were drawn 1–10,000 times using uniform sampling with different seeds, followed in the case of > 1 sample by selection of the sample that best matched the original distribution of variables, judged by statistical comparisons of probability density functions. The sampled data were subjected to projection using PCA and the projection was used to predict the remaining data that had not been sampled from the original data set. The experiments were performed in 20 replicates starting with different seeds. The boxes have been constructed using the minimum, quartiles, median (solid line within the box), and maximum of the means squared reproduction error. The whiskers add 1.5 times the interquartile range (IQR) to the 75th percentile or subtract 1.5 times the IQR from the 25th percentile. The observed single mean squared errors are shown as dots. For a statistical evaluation of the results, see **Table 2**. **Panels A–E** show data sets #2 - #6 described in the methods section and in the left columns of **Table 3**). The figure has been created using the R software package (version 4.0.4 for Linux; https://CRAN.R-project.org/ [2]) and the R libraries "ggplot2" (https://cran.r-project.org/package=ggplot2 [33]) and "ggthemes" (https://cran.r-project.org/package=ggthemes [34]).

terms of the distribution of each variable was not a PCA-specific observation, but similarly observed across the 20 replicates of the experiments when autoencoding neural networks were used to project the down-sampled data and reconstruct the remaining data.

## Discussion

Class-proportional downsampling of data sets by selecting a fraction of the data with uniformly distributed probability, which is a widely used standard method for downsizing data sets, could be improved in terms of the degree to which the selected sample represents the original data set. The hypothesis pursued in this report appeared to be correct, namely that the standard method of drawing an initial random sample from a data set can be improved by selecting a sample that possesses a distribution as similar as possible to the distribution of the original full data set. The sample selected using the criteria of distributional similarity represented the entire sample better than any initial random sample. This was consistently observed when either PCA or an artificial neural network was used to reconstruct the remaining data.

**Table 2. Results of analysis of variance (ANOVA) of the mean squared data reconstruction errors.** The ANOVA was performed the factors "fraction" and "number of trials" of the mean squared errors of the reconstruction of the remaining data from the downsampled data. Depending on the size of the data sets, class-proportional uniformly distributed random samples of, e.g., 0.001, 0.01, 0.1, 1, 5, 10, 25, and 50% of the original data were drawn (Fig 2). The experiments were performed in 20 replicates, each stating at a different and non-redundant seed.

| Data set # | Data set name | ANOVA factor | Degrees of freedom | F value | p-value |
|---|---|---|---|---|---|
| 2 | Iris | Fraction | 4 | 61.809 | $< 2 \cdot 10^{-16}$ |
| | | Number of trials | 4 | 33.867 | $< 2 \cdot 10^{-16}$ |
| | | Fraction * number of trials | 16 | 7.423 | $1.31 \cdot 10^{-15}$ |
| | | Residuals | 475 | | |
| 3 | Artificial | Fraction | 2 | 1653.368 | $< 2 \cdot 10^{-16}$ |
| | | Number of trials | 3 | 10.247 | $2.35 \cdot 10^{-6}$ |
| | | Fraction * number of trials | 6 | 2.732 | 0.0139 |
| | | Residuals | 228 | | |
| 4 | Single Cell | Fraction | 2 | 358.707 | $< 2 \cdot 10^{-16}$ |
| | | Number of trials | 3 | 3.257 | 0.0224 |
| | | Fraction * number of trials | 6 | 2.338 | 0.0328 |
| | | Residuals | 228 | | |
| 5 | FACS | Fraction | 2 | 649.689 | $< 2 \cdot 10^{-16}$ |
| | | Number of trials | 3 | 9.953 | $3.43 \cdot 10^{-6}$ |
| | | Fraction * number of trials | 6 | 4.309 | 0.000387 |
| | | Residuals | 228 | | |
| 6 | miRNA | Fraction | 4 | 1307.21 | $< 2 \cdot 10^{-16}$ |
| | | Number of trials | 4 | 21.81 | $< 2 \cdot 10^{-16}$ |
| | | Fraction * number of trials | 16 | 4.02 | $3.28 \cdot 10^{-7}$ |
| | | Residuals | 475 | | |

A p-value $< 2 \cdot 10^{16}$ is a technical constraint and corresponds to the minimum floating-point number that can be stored by R on standard computers.

This seems to contradict the assumption that uniformly randomly selected subsamples represent the overall data set equally well and that only sample size is responsible for how well the overall data set is represented, although as expected the latter was also observed.

The present proposal assumes that the relevant information about the structure of a data set is contained in the distribution of its variables. However, it was found that the different measures of similarity in the distribution of variables have different properties as criteria for subsample selection. Although the tendency to draw a subsample that reflects well the structure of the original data set improved, the proposed method did not always point at the best subsample in terms of reconstruction of the remaining data, suggesting that the criterion of similarity in the distribution of variables was independent of the subsequent analyses and not the sole criterion for the most representative subsample. In fact, sometimes the first sample was better for reconstructing the rest of the data than the sample selected according to the similarity criterion. Importantly, the trend toward improved reconstruction by applying this criterion was stable. Nevertheless, by applying solely the criterion of similarity of the distribution of the downsampled variables to that of the respective variables in the full data set, the method adheres to the concept of random downsampling and does not anticipate any results of subsequent data analyses performed on the reduced data set.

The overall success in the present hypothesis testing was achieved despite the brute force approach of the improvement of random sampling from a data set cannot be fulfilled on contemporary computers. It is not nearly necessary to test the septillion possible combinations of class-proportional samples from the Iris data set to select a sample that generally provides

**Table 3. Mean squared errors of PCA and autoencoder-based data reproduction of the remaining data from the sampled data subset.** Samples of 0.001 and 0.01%, for the smaller iris and miRNA data sets of 1% and 10%, of the data were drawn once using uniform sampling or 1,000 times using uniform sampling with different seeds, followed by selection of the sample that best matched the original distribution of variables, judged by statistical comparisons of probability density functions. The sampled data were subjected to projection using either PCA or a single-layer autoencoder, and then the projection parameters were used to predict the remaining data that had not been sampled from the original data set. The experiments were performed in 20 replicates starting with different and non-redundant seeds, and the means and standard deviations of the mean square errors of the data reproduction obtained during these replicates are shown.

| Data set | | | | | Downsampling experiments | | | | | | | | | | |
| Number | Name | Size | | | Trials | | | | | Trials | | | | | |
| | | Instances | Features | | Percent sampled | 1 trial PCA | 1,000 trials PCA | 1 trial Autoencoder | 1,000 trials Autoencoder | Percent sampled | 1 trial PCA | 1,000 trials PCA | 1 trial Autoencoder | 1,000 trials Autoencoder | |
|---|---|---|---|---|---|---|---|---|---|---|---|---|---|---|---|
| 2 | Iris | 150 | 4 | | 1 | 0.14180 ± 0.04654 | 0.10003 ± 0.00685 | 0.16102 ± 0.07157 | 0.10049 ± 0.00762 | 10 | 0.09951 ± 0.00901 | 0.09049 ± 0.00217 | 0.07306 ± 0.01271 | 0.06273 ± 0.00528 | |
| 3 | Artificial | 30,000 | 10 | | 0.001 | 154.8562 ± 21.04911 | 132.7860 ± 14.16717 | 291.8664 ± 98.05335 | 238.0392 ± 36.28223 | 0.01 | 111.9968 ± 23.77421 | 96.0451 ± 12.64811 | 279.0497 ± 107.61263 | 228.9243 ± 43.90218 | |
| 4 | Single Cell | 23,377 | 4 | | 0.001 | 2510.473 ± 1215.4342 | 1960.399 ± 767.2990 | 46284.66 ± 10471.239 | 36119.58 ± 11173.769 | 0.01 | 116.858 ± 522.6049 | 208.169 ± 650.8854 | 40136.55 ± 10603.902 | 28433.88 ± 6664.088 | |
| 5 | FACS | 111686 | 6 | | 0.001 | 1.08773 ± 0.34595 | 0.81541 ± 0.10037 | 1.22914 ± 0.65268 | 0.83760 ± 0.17717 | 0.01 | 0.35906 ± 0.13173 | 0.26569 ± 0.02517 | 0.26754 ± 0.07288 | 0.21012 ± 0.03627 | |
| 6 | miRNA | 94 | 184 | | 1 | 1.16598 ± 0.22298 | 0.96829 ± 0.05236 | 3.16437 ± 0.55408 | 2.86462 ± 0.58554 | 10 | 0.82086 ± 0.11161 | 0.73401 ± 0.04832 | 2.83602 ± 0.30254 | 2.69106 ± 0.04607 | |

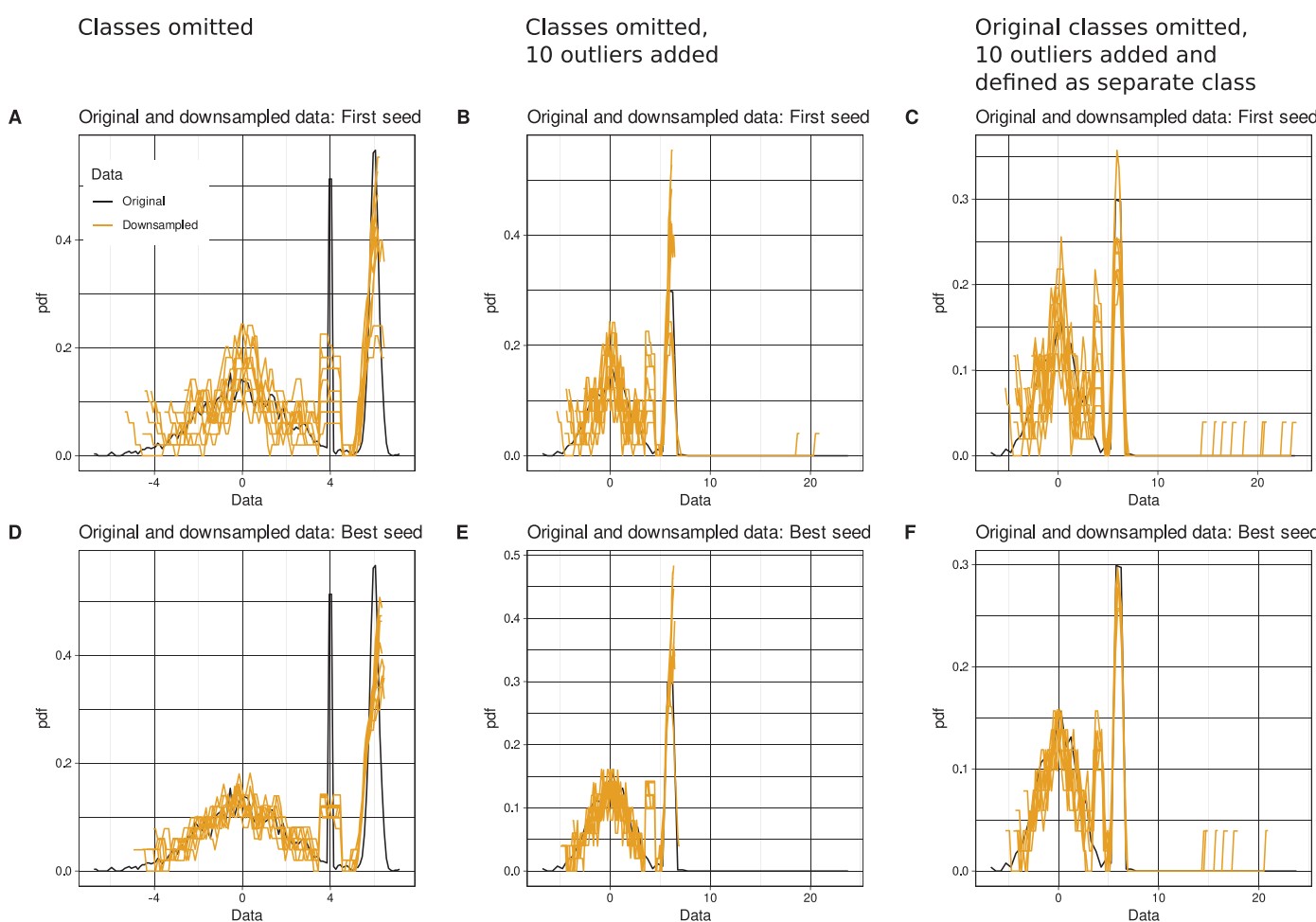

**Fig 3. Special cases of small subgroups or outliers when sampling 50 data points 10 times uniformly from an original artificial data set of 3,000 instances, using either classical uniform sampling or the optimized sampling method.** The original data included 3,000 instances from three classes with prior probabilities of [0.6, 0.1, 0.3]. The distributions of cases within the classes follow Gaussian distributions with means = [0, 4, 6] and standard deviations = [2, 0.001, 0.2]. The panels show the probability density distribution of the original data (black lines) and the sampled subsets (yellow lines). The probability density distribution is estimated using Pareto Density Estimation (PDE [32]). PDE is a kernel density estimator designed to be particularly useful for detecting classes [32]. Three scenarios were tested, presented in columns from left to right. *First* (**panels A and D**), the class information of the Gaussian mixture was made unavailable during downsampling. *Second* (**panels B and E**), 20 outliers were added far to the right of the largest data point of the 3,000 instances, but no class information was provided during downsampling. *Third* (**panels C and F**), the 20 outliers were assigned to an arbitrary class #2, while all the previous 3,000 data points were assigned to class #1. The **upper row** shows 10 downsampled data subsets when using just the first of 10 different seeds. The **lower row** shows 10 optimal downsampled data subsets identified from 1,000,000 non-redundant seeds each. The figure has been created using the R software package (version 4.0.4 for Linux; https://CRAN.R-project.org/ [2]) and the R libraries "ggplot2" (https://cran.r-project.org/package=ggplot2 [33]), "ggthemes" (https://cran.r-project.org/package=ggthemes [34]) and "AdaptGauss" (https://cran.r-project.org/package=AdaptGauss [35]).

better information about the entire data set than an initial random sample often does. Therefore, despite the fact that its brute force approach cannot be satisfied in most cases, it seems justified to propose the present method as an optimized approach for class-proportional downsampling.

It should be noted that this proposal does not fundamentally challenge the standard approach of uniform random sampling from a larger data set. This includes the condition that downsampling very small subsets cannot be always consistent. The portion downsampled from the data must be large enough to yield statistically relevant sample sizes of subgroups. If this is not true, another downsampling method with selective filtering and subgroup enrichment, etc., should be chosen instead of the present method. The approach presented here

maintains the probability with which instances of a data set become part of the sample from uniform downsampling. Indeed, any sample selected on the basis of distributional similarity could also have been selected in a standard random sampling procedure. Therefore, the selection procedure is unlikely to cause disruptive changes in subsequent analyses of the downsampled data set.

A special case is small classes or outliers in a data set. If the class information is present, the method samples the defined fraction of the data from each class, as demonstrated with data set #1 that contains a moderately small class of 10% of the cases. However, prior class information is often incomplete or non-existent, while it is only the task of future data analysis with unsupervised methods to detect a class structure in a data set. To further discuss the behavior of the proposed downsampling method in such settings, the resampling experiment of data set #1 was repeated with the class information omitted, i.e., all instances were assigned to a single class #1. The results indicated that small hidden classes were likely to be adequately represented in the downsampled data (**Fig 3A** and **3D**). In another experiment, outliers were added as 10 consecutive integer numbers starting at x = 15, which were far to the right of the maximum of the data at x = 7.169319. Application of the downsampling algorithm resulted in consistent omission of these added outliers (**Fig 3E**), while in the experiments where the first random sample was always taken without further control, some outliers were captured (**Fig 3B**). This could be reproduced with the advanced downsampling method when the outliers were assigned to a separate class (**Fig 3C** and **3F**). So, if outliers are suspected, this procedure can be used. Of course, the researcher must choose a valid outlier definition, which might be only possible after adequate data transformation. However, these considerations are standard in data science and are not part of the present proposal for improving uniform downsampling. In fact, if downsampling and especially uniform downsampling, i.e., every data point has the same chance of being drawn, is contraindicated, the presented method is obviously not applicable either. The proposed method is best suited for independent, numerical, tabular data. For sequential data as an example, such as continuous Markov processes, reducing the number of data points can lead to information loss, since the respective values depend on their predecessors. In addition, the R implementation provides feature selection via PCA with the idea that only relevant features should contribute to the selection of a representative data subset among many possible uniformly drawn subsets. If there is doubt that PCA projection provides an adequate perspective on the data set to select its relevant variables, but alternatives such as independent component analysis or other feature selection methods are preferable, it is recommended that feature selection should be addressed only at a later stage of data analysis; therefore, the library's feature is disabled by default. In addition, applying complex statistical procedures to the entire data set could run counter to the goal of the present method to shrink a data set before any further data processing because its size exceeds computer capacity. In the present experiments, the use of feature selection produced no clear improvements, so the corresponding tests are not reported in detail.

## Conclusions

The enormous sizes of nowadays biomedical data sets push the hardware equipment of current computers to their limit, even for seemingly standard analytical tasks such as data projection or clustering. Typically, the $O(n^2)$ size of distance matrices exceeds the memory of most, even large, computers. Reducing large biomedical data by downsampling is therefore a common early step in their processing. The presented method offers an improvement over the uniform, random, class-proportional downsampling, which is standard and implemented in all software packages, to obtain data sets of manageable size. Optimized distribution-preserving

downsampling favors the selection of samples which represent the original data set significantly better than if just the first randomly drawn sample is chosen by default. The experiments in this report have shown that the proposed method yields data subsets from which the structure of the entire data can be better reconstructed compared to the standard method in particular when projection methods are used for further processing of the data. The fidelity was dependent on both, the number of cases drawn from the original, and the number of samples drawn from which the best one could be selected, the latter confirming the suitability of the chosen approach to select a representative subsample from a larger data set. Since similarity of distribution is used as the only selection criterion, the proposed method does not in any way affect the results of a later planned analysis or bias the representative subsample toward the expected results.

## Supporting information

**S1 File.** The supplementary information comprises (i) probabilistics of the possible combinations in class-proportional downsampling scenarios and (ii) the detailed description of the PCA based reconstruction of data including relevant equations.
(PDF)

## Author Contributions

**Conceptualization:** Jörn Lötsch, Alfred Ultsch.

**Data curation:** Jörn Lötsch.

**Formal analysis:** Jörn Lötsch.

**Funding acquisition:** Jörn Lötsch.

**Investigation:** Jörn Lötsch.

**Methodology:** Jörn Lötsch.

**Project administration:** Jörn Lötsch.

**Resources:** Jörn Lötsch.

**Software:** Jörn Lötsch, Sebastian Malkusch.

**Supervision:** Jörn Lötsch, Sebastian Malkusch.

**Validation:** Jörn Lötsch, Sebastian Malkusch, Alfred Ultsch.

**Visualization:** Jörn Lötsch.

**Writing – original draft:** Jörn Lötsch, Alfred Ultsch.

**Writing – review & editing:** Jörn Lötsch, Sebastian Malkusch, Alfred Ultsch.

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
