## [Decision Letter · Decision Letter 0]

1 Jun 2021

PONE-D-21-09268

Optimal distribution-preserving downsampling of large biomedical data sets (opdisDownsampling)

PLOS ONE

Dear Dr. Lötsch,

Thank you for submitting your manuscript to PLOS ONE. After careful consideration, we feel that it has merit but does not fully meet PLOS ONE’s publication criteria as it currently stands. Therefore, we invite you to submit a revised version of the manuscript that addresses the points raised during the review process.

We look forward to receiving your revised manuscript.

Kind regards,

Saeed Mian Qaisar, Ph.D.

Academic Editor

PLOS ONE

Journal Requirements:

We note that you have stated that you will provide repository information for your data at acceptance. Should your manuscript be accepted for publication, we will hold it until you provide the relevant accession numbers or DOIs necessary to access your data. If you wish to make changes to your Data Availability statement, please describe these changes in your cover letter and we will update your Data Availability statement to reflect the information you provide.

Please remove your figures from within your manuscript file, leaving only the individual TIFF/EPS image files, uploaded separately.  These will be automatically included in the reviewers’ PDF.

Additional Editor Comments:

Dear Authors,

Reviewers have now commented on your paper. They are advising that you revise your manuscript. If you are prepared to undertake the work required, I would be please to reconsider my decision.

The reviewer comments can be found at the end of this email or can be accessed online.

While revising your paper please also consider the following points.

1. It is recommended to add a work flow diagram in the beginning of section "Methods". It shows different processing stages in the system.

2. It is recommended that authors submit the studied datasets as supplementary zip file in the .csv OR .xls formats while submitting the revised version of paper.

Reviewers' comments:

Reviewer's Responses to Questions

**Comments to the Author**

1. Is the manuscript technically sound, and do the data support the conclusions?

Reviewer #1: Yes

2. Has the statistical analysis been performed appropriately and rigorously? 

Reviewer #1: Yes

3. Have the authors made all data underlying the findings in their manuscript fully available?

Reviewer #1: Yes

4. Is the manuscript presented in an intelligible fashion and written in standard English?

Reviewer #1: Yes

5. Review Comments to the Author

Reviewer #1: The authors present their proposal of a downsampling method of large biomedical data sets. The method, experiments and results are described clearly.

I have few comments and questions that should be answered in the revised version. The used datasets are rather similar concerning number of features used and also basic characteristics of individual examples.

How would the method work for a larger number of features (e.g. more than 50)?

Is feature selection necessary before starting the method?

What procedure would the authors recommend for outlier handling? In many cases outliers represent rare cases and not noise. Thus they should not be removed from the data set.

It would be welcome if the authors formulate a recommendation, for which type of data the proposed method is suitable.

Is it applicable to time series or signals in which we search for certain patterns?

6. PLOS authors have the option to publish the peer review history of their article (what does this mean?). If published, this will include your full peer review and any attached files.

Reviewer #1: No

---

## [Author Response · Author response to Decision Letter 0]

23 Jun 2021

Dear Professor Qaisar,

Thank you very much for having handled our manuscript PONE-D-21-09268 entitled “Optimal distribution-preserving downsampling of large biomedical data sets (opdisDownsampling)" and considering it for publication in PLoS One after minor revision. 

We would like to thank the reviewers for their helpful comments, which we have addressed as follows. Specifically, the reviewers’ comments are given in black Calibri font and our responses are given below each point in red Times Roman font.

Comments to the Author

Reviewer #1: The authors present their proposal of a downsampling method of large biomedical data sets. The method, experiments and results are described clearly.

I have few comments and questions that should be answered in the revised version. The used datasets are rather similar concerning number of features used and also basic characteristics of individual examples.

How would the method work for a larger number of features (e.g. more than 50)?

We have added a higher dimensional data set with d = 184 features as data set #6, described in the methods section as “The sixth data set was again of biomedical origin and consisted of a "miRNA and chronic pain" data set containing measurements of 184 microRNAs in 94 instances grouped into d = 5 classes of sizes [14, 24, 18, 17, 21]. The data set is freely available under the Creative Commons license CC By 4.0 at https://data.mendeley.com/datasets/37fnjc4yhm/2 (downloaded at June 14, 2021). Of its two versions, the "Profiling raw data.xlsx" file dated February 19, 2020 was selected [18].”

Is feature selection necessary before starting the method?

We would like to thank the reviewer for this comment. At an earlier stage of the project work, we had thought along the same lines and implemented a feature selection option in the R library. In later stages, we had hesitated to implement it in the final version and had removed it. The reviewer's comment encouraged us to add it back as an option in the library, and so we are grateful for that comment. The option is now described in the "Implementation" chapter of the "Methods" section as “Optionally, the library also provides (vii) a fast PCA-based feature selection ("PCAimportance") to exclude variables from the assessment of similarity to the distributions of the downsampled data and the original data to avoid unnecessarily optimizing the sample for irrelevant variables. This feature is disabled by default. The standard "prcomp" method implemented in base R [2] is used, and variables’ selection was based the loadings on relevant principal components according to the Kaiser-Gutman criterion, i.e., on PCs with eigenvalues > 1. Specifically, the relevant variables are selected as suggested in the R library "factoextra" (https://cran.r-project.org/package=factoextra [29, 30]) based on the expected value if the contributions were uniform, which is given as 100/length(contrib) with "contrib" denoting the list of contribution magnitudes of each variable to a given PC.”. In the discussion, we comment on this option, also including the reasons why we hesitated first to include it: “In addition, the R implementation provides feature selection via PCA with the idea that only relevant features should contribute to the selection of a representative data subset among many possible uniformly drawn subsets. If there is doubt that PCA projection provides an adequate perspective on the data set to select its relevant variables, but alternatives such as independent component analysis or other feature selection methods are preferable, it is recommended that feature selection should be addressed only at a later stage of data analysis; therefore, the library’s feature is disabled by default. In addition, applying complex statistical procedures to the entire data set could run counter to the goal of the present method to shrink a data set before any further data processing because its size exceeds computer capacity. In the present experiments, the use of feature selection produced no clear improvements, so the corresponding tests are not reported in detail.”

A new version of the R library including this feature has been uploaded to CRAN.

What procedure would the authors recommend for outlier handling? In many cases outliers represent rare cases and not noise. Thus they should not be removed from the data set.

We thank the reviewer for the comment, as it points out a detail to which we had not given the necessary attention. Since we were asked to make a minor revision, we felt it was inappropriate to change the structure of the report and have included the response to the reviewer's question in the discussion, including the insertion of an additional Figure 3: “A special case is small classes or outliers in a data set. If the class information is present, the method samples the defined fraction of the data from each class, as demonstrated with data set #1 that contains a moderately small class of 10 % of the cases. However, prior class information is often incomplete or non-existent, while it is only the task of future data analysis with unsupervised methods to detect a class structure in a data set. To further discuss the behavior of the proposed downsampling method in such settings, the resampling experiment of data set #1 was repeated with the class information omitted, i.e., all instances were assigned to a single class #1. The results indicated that small hidden classes were likely to be adequately represented in the downsampled data (Figure 3 A and D). In another experiment, outliers were added as 10 consecutive integer numbers starting at x = 15, which were far to the right of the maximum of the data at x = 7.169319. Application of the downsampling algorithm resulted in consistent omission of these added outliers (Figure 3 E), while in the experiments where the first random sample was always taken without further control, some outliers were captured (Figure 3 B). This could be reproduced with the advanced downsampling method when the outliers were assigned to a separate class (Figure 3 C and F). So, if outliers are suspected, this procedure can be used. Of course, the researcher must choose a valid outlier definition, which might be only possible after adequate data transformation. However, these considerations are standard in data science and are not part of the present proposal for improving uniform downsampling.”

It would be welcome if the authors formulate a recommendation, for which type of data the proposed method is suitable. Is it applicable to time series or signals in which we search for certain patterns?

We do not think that time series should be downsampled using our method. To respond to the reviewer’s questions, we have added to the discussion: “However, these considerations are standard in data science and are not part of the present proposal for improving uniform downsampling. In fact, if downsampling and especially uniform downsampling, i.e., every data point has the same chance of being drawn, is contraindicated, the presented method is obviously not applicable either. The proposed method is best suited for independent, numerical, tabular data. For sequential data as an example, such as continuous Markov processes, reducing the number of data points can lead to information loss, since the respective values depend on their predecessors.”

Again, we would like to thank you for having processed the manuscript and you and the reviewers for many helpful suggestions to improve it. We hope that the manuscript now meets the criteria for publication in PLoS One and are very much looking forward to hearing back from you.

Sincerely yours

Jörn Lötsch and Sebastian Malkusch

---

## [Decision Letter · Decision Letter 1]

26 Jul 2021

Optimal distribution-preserving downsampling of large biomedical data sets (opdisDownsampling)

PONE-D-21-09268R1

Dear Dr. Lötsch,

We’re pleased to inform you that your manuscript has been judged scientifically suitable for publication and will be formally accepted for publication once it meets all outstanding technical requirements.

Kind regards,

Saeed Mian Qaisar, Ph.D.

Academic Editor

PLOS ONE

Additional Editor Comments (optional):

Dear Authors,

I am pleased to tell you that your work has now been accepted for publication in the PLOS ONE Journal.

Thank you for submitting your work to this journal.

Reviewers' comments:

Reviewer's Responses to Questions

**Comments to the Author**

1. If the authors have adequately addressed your comments raised in a previous round of review and you feel that this manuscript is now acceptable for publication, you may indicate that here to bypass the “Comments to the Author” section, enter your conflict of interest statement in the “Confidential to Editor” section, and submit your "Accept" recommendation.

Reviewer #1: All comments have been addressed

2. Is the manuscript technically sound, and do the data support the conclusions?

Reviewer #1: Yes

3. Has the statistical analysis been performed appropriately and rigorously? 

Reviewer #1: Yes

4. Have the authors made all data underlying the findings in their manuscript fully available?

Reviewer #1: Yes

5. Is the manuscript presented in an intelligible fashion and written in standard English?

Reviewer #1: Yes

6. Review Comments to the Author

Reviewer #1: The manuscript has been significantly improved and all questions and comments adequately addressed. I have no additional comments or suggestions.

7. PLOS authors have the option to publish the peer review history of their article (what does this mean?). If published, this will include your full peer review and any attached files.

Reviewer #1: No

---

## [Editor Report · Acceptance letter]

28 Jul 2021

PONE-D-21-09268R1 

Optimal distribution-preserving downsampling of large biomedical data sets (opdisDownsampling) 

Dear Dr. Lötsch:

I'm pleased to inform you that your manuscript has been deemed suitable for publication in PLOS ONE. Congratulations! Your manuscript is now with our production department. 

Kind regards, 

on behalf of

Dr. Saeed Mian Qaisar 

Academic Editor

PLOS ONE